# Targeting the IL-23 Receptor Gene: A Promising Approach in Inflammatory Bowel Disease Treatment

**DOI:** 10.3390/ijms26104775

**Published:** 2025-05-16

**Authors:** Ploutarchos Pastras, Ioanna Aggeletopoulou, Konstantinos Papantoniou, Christos Triantos

**Affiliations:** Division of Gastroenterology, Department of Internal Medicine, University of Patras, 26504 Patras, Greece; ploutarchosp96@gmail.com (P.P.); g.papanton@yahoo.gr (K.P.); chtriantos@upatras.gr (C.T.)

**Keywords:** IL-23 receptor, inflammatory bowel disease, variant, gene, treatment

## Abstract

Inflammatory bowel disease (IBD), which includes Crohn’s Disease (CD) and ulcerative colitis (UC), is characterized by chronic inflammation of the gastrointestinal tract. A key component of the inflammatory pathway in IBD is interleukin 23 (IL-23), which promotes the differentiation and maintenance of Th17 cells. These cells are major contributors to intestinal inflammation and the release of pro-inflammatory cytokines. A dysregulated IL-23/Th17 axis can lead to excessive gut inflammation. Notably, IL-23 affects Th17 cell responses differently in UC and CD, fostering IL-17 production in UC and interferon-gamma (IFN-γ) production in CD. Genetic studies have pinpointed specific variants of the IL-23 receptor (IL23R) gene that confer protection against IBD. The *R381Q* (rs11209026) variant has been linked to a reduced risk of developing both CD and UC. Additionally, other variants, such as *G149R* (rs76418789) and *V362I* (rs41313262), inhibit IL23R function by disrupting intracellular trafficking and protein stability. This disruption results in decreased phosphorylation of downstream signal transducers, such as STAT3 and STAT4, and reduced IL23R expression on the cell surface, ultimately dampening the activation of pro-inflammatory pathways. The protective effects of these genetic variants underscore the IL-23/IL23R pathway as a significant therapeutic target in IBD management. Therapies designed to modulate this pathway have the potential to reduce pro-inflammatory cytokine production and enhance anti-inflammatory mechanisms. Ongoing research into the IL23R gene and its variants continues to provide valuable insights, paving the way for more targeted and effective treatments for IBD patients.

## 1. Introduction

The function of the gastrointestinal tract plays a crucial role in maintaining immune homeostasis. This tract serves a dual role: it preserves tolerance to food antigens, commensal bacteria, and the body’s own antigens, while also triggering an inflammatory response to defend against pathogens [1]. An imbalance between these two functions can lead to gastrointestinal disorders such as inflammatory bowel disease (IBD) [1]. Ulcerative colitis (UC) and Crohn’s disease (CD) are the main clinical forms of IBD, sharing both similarities and differences in their presentation [2]. Despite these differences in pathophysiology, UC and CD have common pathogenetic mechanisms that justify their classification under the umbrella of IBD [3].

The main characteristic of IBD is the lifelong periods of relapse and remission, which are characterized by mucosal inflammation resulting from an inappropriate innate and adaptive immune response [4,5,6]. The interaction between abnormal immune regulation, gut microbiota, environmental factors, and the impaired integrity of the gut barrier promotes the development and progression of IBD [7,8]. However, the pathophysiology of the disease is not fully understood, with several pathogenetic pathways yet to be elucidated [9].

The incidence of IBD is rising in the general population, and its current treatment landscape is multifaceted, complex, and personalized, as available medications address only specific aspects of the disease’s pathophysiology [10]. Consequently, numerous studies have been conducted to investigate the unexplored pathophysiological mechanisms of IBD, with particular emphasis on genetic susceptibility [11]. Several shared genetic loci have been identified between UC and CD, suggesting that treatments targeting these loci could benefit patients across the IBD spectrum [12].

The interleukin-23 receptor (IL23R) gene is among the most promising genetic targets. Extensive genome-wide association studies (GWAS) have revealed that variants in IL23R modify susceptibility to both UC and CD, indicating the vital role of interleukin (IL)-23 signaling in IBD pathogenesis [13]. Building on this genetic insight, this review explores the pathophysiological association between the IL23R gene and IBD, and discusses how this connection may guide the development of targeted therapies. We also present current therapeutic strategies that aim to modulate the IL-23/IL-23R pathway, offering new possibilities for personalized treatment in IBD.

## 2. The Role of IL-23 and IL-23R in IBD

### 2.1. IL-23 and IL-23R

IL-23 is a heterodimeric member of the IL-12 family. The IL-12p40 subunit is common to both IL-12 and IL-23, while the other subunits are IL-23p19 for IL-23 and IL-12p35 for IL-12 [14]. IL-23 and IL-12 are cytokines associated with pro-inflammatory reactions, whereas other members of the IL-12 family (IL-27 and IL-35) play an inhibitory role [15]. The IL-23 receptor is a heterodimeric protein composed of two subunits: IL-12Rb1 and IL-23R. The IL-12Rb1 subunit is shared by both the IL-12 receptor and the IL-23 receptor, whereas the IL-23R subunit specifically binds to IL-23p19 of IL-23 [16]. To distinguish the IL-23R heterodimer receptor from the individual subunit chains, in this manuscript, the distinct IL-23 subunit will be referred to as IL23Ra, and the IL-23R heterodimer receptor will be labeled IL23R.

### 2.2. IL-23 Production

IL-23 production is affected by various factors. Antigen-presenting cells (APCs), particularly macrophages and dendritic cells (DCs), are the primary cellular sources of IL-23 production [17]. These cells increase the creation of IL-23 in the intestine when exposed to gut microbial stimuli. Response to *Escherichia coli* and *Enterococcus faecalis* stimulates IL-23 secretion by lamina propria CD14+ CD33+ macrophages [18]. In the presence of intestinal microbiota, CD11c + DCs in the terminal ileum lamina propria constitutively express the IL-23p40 subunit, indicating that IL-23 expression is greater in the healthy mucosa of this region [19]. In the inflamed mucosa of both CD and UC patients, the IL23-p19 subunit is present at increased levels, produced mainly from DCs and CD68+ macrophages [20]. For example, in CD, the severity of macroscopic lesions at endoscopy is positively correlated with IL-23p19 mRNA levels [21]. In CD refractory to TNF antagonism, the IL-23p19 levels in intestinal lamina propria CD14+ macrophages are augmented due to the accumulation of resistance to apoptosis, TNFR2+ IL23+ CD4+ T cells [22].

Molecular mechanisms and cellular signaling play a key role in IL-23 production by APCs. More specifically, when microbial agonist lipopolysaccharide (LPS) binds to Toll-like receptor 4 (TLR4), it promotes the production of IL-12 and IL-23 [23], while bacterial peptidoglycan in vitro stimulates the production of the IL-23p19 subunit over the IL12-p35 subunit [24]. Monocyte-delivered dendritic cells (MoDCs) promote IL-23 in response to both intact Gram-negative and Gram-positive bacteria, in contrast to the activation of DCs via TLR ligands [25]. Additionally, stimulation of cytosolic nucleotide binding oligomerization domain-containing protein 2 (NOD2) receptor in DCs and CD4+ T cells induces the production of IL-23 [26]. There are NOD2 disease-related variants that augment IL-23p19 secretion from DCs through decreased expression of the microRNA (miR-29) when DCs are stimulated by *Escherichia coli* [27]. Two additional molecular mechanisms contributing to IL-23 production involve extracellular adenosine nucleotides and intracellular metabolism within DCs and macrophages [28,29]. Extracellular adenosine increases IL-23p19 levels through the purinergic receptor P2 when DCs are exposed to *Escherichia coli* [17]. Macrophage metabolism also affects IL-23 production, as IBD-linked genes encode proteins like the enzyme FAMIN (a purine nucleotide recycling enzyme in macrophages) [30].

It is also crucial to consider factors other than microbial stimulation that affect IL-23 production. IL-10 reduces production via paracrine signaling. Mutations in the IL-10 receptor reduce the expression of IL-23p19 and lead to uncontrolled inflammation [31]. In addition, ATP and IL-1β induce the production of IL-23, and IFN-γ amplifies its creation [32]. Neutrophils are also a significant source of IL-23 in the gut via CXCR1/CXCR2 receptors, which regulate IL-23 through chemokines of the CXCL8 family [33]. A schematic overview of these mechanisms is illustrated in Figure 1, highlighting the interactions between IL-23, immune cell subsets, microbial stimuli, and inflammatory mediators.

### 2.3. IL-23 and Its Multifaceted Role in IBD Pathogenesis

IL-23 plays a crucial role in IBD by activating various immune cell types. Its main influence is on Th17 cells, which are located in the intestinal lamina propria and induce the IL-17A production [34]. Th17 cells are largely produced in mice with specific Gram-positive bacteria in their microbiota [35], and the mucosa of IBD-inflamed patients contains more Th17 cells than the mucosa of healthy patients [36]. IL23 does not act alone; it requires IL-6, IL-21, and transforming growth factor β (TGFβ) to differentiate naïve CD4+ T cells into Th17 cells [37]. A combination of these interleukins with low concentrations of TGFβ in vitro enhances the expression of retinoid acid receptor-related orphan receptor γt (RORγt), suppresses the transcription factor Forkhead box P3 (FoxP3), and upregulates IL-23R expression [38]. The interaction between IL-23 and IL-23R promotes further IL-17 production [39], highlighting IL-23’s role in the survival and expansion of Th17 cells, as naïve CD4+ T cells do not express IL-23R [40].

In addition to IL-17, Th17 cells produce granulocyte-macrophage colony-stimulating factor (GM-CSF) and IL-12 when IL-23 is present [41,42]. Additionally, IL-12 and IL-23 can restimulate Th17-polarized CD4^+^ T cells in vitro, resulting in a decrease in IL-17A and IL-17F and an increase in interferon-gamma (IFNγ) expression when TGFβ is absent [43]. This Th17 phenotype is similar to Th1 cells, indicating the heterogeneity and plasticity of Th17 cells under different circumstances [44]. This heterogeneity could explain the controversial role of IL-17 in IBD development, and the different results observed in UC and CD. A neutralizing human anti-IL-17A antibody (secukinumab) has shown no results in moderate to severe CD, while a human anti-IL17 receptor A monoclonal antibody (brodalumab) improved active disease in CD in a randomized, double-blind clinical trial [45,46]. However, a humanized anti-interferon gamma antibody (fontolizumab) demonstrated clinical activity and safety in patients with moderate to severe CD in a multicenter phase 2 study involving 133 patients [47]. As for UC, a recent study indicated that suppression of the IL-17 signaling pathway may alleviate the progression of UC [48]. Regardless of the differences between disease types, the IL23/Th17 axis can induce inflammatory pathways in IBD. For instance, transferring naïve CD4^+^ T cells deficient in IL23R expression into mice reduced IL-17^+^ IFNγ^+^ cells in the colon [49], and IFNγ^+^ Il-17^+^ CD4^+^ T cells have been observed in the inflamed mucosa of both CD and UC patients but not in controls [50].

Furthermore, IL-23 affects IBD not only through its action on Th17 cells but also by constraining regulatory CD4^+^ T cells (Tregs) in the intestine. Tregs suppress IL-23 production by intestinal macrophages [49,51]. Tregs express anti-inflammatory cytokines (IL10 and IL-35), TGFβ, and the transcription factor FoxP3 [52]. In experimental models, the transfer of Treg-enriched cells has been shown to prevent colitis [53], and transferring CD4^+^CD25^+^ Tregs reduces established colitis [54], demonstrating their anti-inflammatory action. IL-23 inhibits the induction of FoxP3^+^ CD4^+^ Tregs in the intestine, blocking the anti-inflammatory effects of Tregs [55]. A recent animal study showed that IL23R signaling impairs the stability and function of Tregs, leading to increased inflammation in IBD patients [56].

IL-23 also significantly influences the activity of innate lymphoid cells (ILC3s). IL-23 stimulation induces ILC3s to produce high levels of IL-17 and IL-22 through STAT3 and STAT5 activation [57,58,59]. Inflamed ileum and colon tissues from CD patients have increased ILC3s responsive to IL-23 compared to healthy, non-inflamed mucosa [60]. Furthermore, an animal study showed that IL-23R on ILCs promotes colitis via IL-22 [61].

In addition to ILC3s, IL-22 is produced by Th17 cells, neutrophilic granulocytes, and γδ T cells in response to IL-23 [62]. Its role in intestinal inflammation is both protective and harmful. On one hand, the lack of IL-22 exacerbates colitis and T cell-mediated colitis in adoptive transfer models, and is necessary for optimal host defense against pathogens [63,64]. On the other hand, IL-22 worsens intestinal inflammation by enhancing endoplasm reticulum (ER) stress in epithelial cells [65,66]. IL-22 inactivation in IBD mouse models significantly reduced colitis levels, correlating with diminished ER stress in the intestinal epithelium [65]. Moreover, IL-22 promotes excessive activation of the cGAS-STING pathway in intestinal mucosa in the absence of the ATG16L1 gene (linked to CD susceptibility), resulting in cell death [66].

Finally, IL-23 influences myeloid cells in several ways. It induces neutrophil production, which may exacerbate established intestinal inflammation [67,68]. In experimental T cell transfer colitis, IL-23 amplifies GM-CSF production by Th17 cells, enhancing extramedullary hematopoiesis, accumulating progenitor granulocyte-monocyte cells in the colon, and activating eosinophils that contribute to tissue damage in colitis [68,69]. It is also important to mention the autocrine effects of IL-23. Some myeloid cells, like macrophages, express IL-23R on their surface. The recruitment of signaling intermediates from the IL-23R complex, such as IL-12Rβ1 and JAK2, along with the endocytic recycling of IL-23R, contributes to the release of pro-inflammatory cytokines by human macrophages, stimulated by IL-23 [70]. Figure 2 illustrates the multifaceted role of IL-23 in IBD pathogenesis, highlighting its interactions with adaptive and innate immune cells and the downstream inflammatory pathways involved.

## 3. IL-23R Gene in IBD

IL-23R has been shown to influence chronic inflammatory diseases through its expression on T cells and other immune cells in specific individuals [71]. When IL-23 binds to IL23R, Janus kinase 2 (JAK2) is activated via the IL23Ra subunit, leading to subsequent phosphorylation of IL-23R. This is followed by the recruitment and phosphorylation of the signal transducers STAT3 and STAT4 [72]. The phosphorylated STAT3 and STAT4 form homodimers that translocate to the nucleus, where they promote the transcription of pro-inflammatory cytokines in cells expressing IL-23R [73,74]. This transcriptional activity may lead to chronic inflammation through continuous IL-23/IL-23R interaction [75,76]. This signaling pathway is present in IBD, as well as in other chronic inflammatory diseases [77]. Various genome-wide association studies have demonstrated the role of the IL23/IL23R pathway, reporting that variants in the IL-23R gene (located on chromosome 1p31.3) are associated with IBD, mainly playing a protective role by modulating the activity of these signal transducers [78,79]. These variants typically affect the IL-23Ra expression, although they are located in different regions of the IL-23Ra structure. Thus, despite variations in the mutation mechanisms, the functional consequences of these variants are often similar [80].

### 3.1. R381Q (Arg381Gln, rs11209026)

The R381Q variant (rs11209026) is the most extensively studied protective allele of the IL-23R gene, influencing the IBD pathophysiology [70]. A recent study revealed that this allele was found in ancient European genomes dating back over 14,000 years and may have contributed to increased survival by reducing intestinal inflammation and microbiome dysbiosis [81]. Today, its frequency ranges from 0% to 17%, depending on the population, with approximately 5% observed in Europe [81]. A meta-analysis reported that the rs11209026 polymorphism may be protective against developing both CD and UC [82]. While this allele shows association with UC risk in Caucasian populations, no such association was found in Asian populations [82]. This may reflect either the genetic heterogeneity in different regional and ethnic populations or the limited number of studies in Asia [82].

At the DNA level, the functional single nucleotide polymorphism (SNP) involves a guanine (G) to adenine (A) substitution, resulting in an amino acid change in the protein from arginine (R) to glutamine (Q) at position 381 (R381Q) within the cytoplasmic domain of IL-23R [83]. Owing to its position, this polymorphism potentially affects the surface localization of the receptor and signal transduction [84]. Although rs11209026 is a non-synonymous coding variant located in exon nine, encoding the intracellular tail of the IL-23R, it plays a protective role against IBD susceptibility [85]. The variant location is between the transmembrane domain and the putative JAK2 binding site in the cytoplasmic region of the receptor, potentially reducing protein stability [85]. The replacement of the highly conserved Arg381 with Gln381 can alter interactions between IL-23R and its associated JAK2 kinase, decreasing the cellular response to IL-23. This leads to functional changes in the IL-23R signaling pathway, exerting a protective effect in IBD pathogenesis [85].

At the cellular level, rs11209026 decreases IL-23-dependent IL-17 production. The disease-protective Arg381Gln IL-23R variant leads to a loss of function in primary human CD4^+^ and CD8^+^ T cells, resulting in decreased cytokine production through the IL-23/Th17 pathway and enhanced microbial clearance [86]. This phenomenon occurs by a decrease in IL-23-induced STAT3 phosphorylation, resulting in lower secretion of pro-inflammatory cytokines, such as IL-17 and IL-22 in the gut by modulating the duration of the host response [87]. The impaired association between JAK2 proteins and the receptor’s cytoplasmic tail leads to reduced capacity of IL-23R to activate STAT proteins [88]. Thus, the IL-23R rs11209026 variant does not interfere with the Th17 differentiation but attenuates the IL-23-induced Th17 effector functions, specifically IL-17A production [89]. Moreover, this allele correlated to a lower percentage of circulating Th17 cells [90].

Additionally, this SNP exhibits alternative mechanisms of function. Rs11209026 alters IL-23Rα mRNA splicing by decreasing the binding of the splicing enhancer SF2, leading to exon nine skipping. This results in the expression of a soluble IL-23R-encoding mRNA species that dampens the IL-23R signaling and decreases the ability of the host to generate a Th17 phenotype in response to IL-23 stimulation [91]. A study also indicated that this SNP decreases IL-23 receptor recycling and impairs its assembly with Janus kinase and STAT pathway components, thereby modulating macrophage-mediated inflammatory pathways [70].

### 3.2. G149R (Gly149Arg, rs76418789)

The G149R variant (rs76418789) has been extensively studied in Japan [92], Korea [93], and China [94], as this allele is rare among Europeans and Africans but more prevalent in Asia (approximately ten times greater in East Asians than in Europeans—3.7% vs. 0.34%, respectively) [95]. A recent case–control study in Japan showed a lower risk of UC in patients with the G149R variant [96]. Despite its protective role in IBD, this allele is rare among the general population, with a frequency of less than 2% in healthy individuals [45].

This mutation consists of a functional SNP resulting in an amino acid substitution from glycine (G) to arginine (R) (Gly149Arg) at position 149 (G149R) in exon 4, a highly conserved residue located in the extracellular region of the receptor [97]. The G149R variant is retained in the endoplasmic reticulum (ER) as unfolded polypeptides due to impaired receptor maturation, preventing simultaneous trafficking from the ER to the Golgi apparatus [80]. This phenomenon is also observed in other diseases involving missense mutations, such as those affecting very low-density lipoprotein receptors, cystic fibrosis transmembrane receptors, and amyloid precursor proteins [98,99,100]. However, in this variant, the stability of the protein is not reduced [80]. A potential explanation for the primary retention of the receptor in the ER is that the variant receptor may be cleaved at the extracellular region by convertases in the Golgi, leading to an increase in receptor degradation and the formation of truncated products [101,102]. The effect of rs76418789 on the IL23R signaling, and consequently on IBD pathophysiology, is similar to that of rs11209026. Specifically, IL-23R activation leads to reduced phosphorylation of STAT3 and STAT4 due to lower receptor expression levels on the cell surface [80].

### 3.3. V362I (Val362Ile, rs41313262)

The V362l variant (rs41313262) is another protective variant of IL-23R for IBD [80]. The epidemiological patterns of this variant in IBD remain poorly characterized, with only limited studies available to date. Notably, the V362I variant showed no association with IBD susceptibility in a Chinese Han population cohort [103]. This mutation consists of a functional SNP that changes an amino acid from valine (V) to isoleucine (I) (Val362Ile) at position 362 in the transmembrane region of the receptor [97]. This variant was identified through targeted re-sequencing, similar to G149R [104]. One of the characteristics of V362I is that it does not exhibit a reduction in the mature/immature receptor ratio, in contrast to R3381Q and G149R. Additionally, it presents a decreased half-life and exhibits lower levels of endogenous receptor expression in V362I homozygous cells [80]. Consequently, it may be characterized by reduced protein stability, similar to R381Q and in contrast to G149R. Furthermore, protein instability may occur due to the location of the V362I substitution. The extra bulk of a methyl group added to the side chain in the transmembrane region of the receptor may disrupt shape complementarity and destabilize the interaction between IL23Ra and IL12Rb1, while still preserving peptide chain rigidity due to branching at the β-carbon, which seems necessary for other transmembrane protein-to-protein interactions [105]. V362I plays a protective role in IBD through the IL-23R signaling pathway in the same manner as G149R [80].

### 3.4. Other IL-23R-Related Gene Variants

Except for rs11209026, rs76418789, and rs41313262, several other SNPs in the IL-23R gene that may influence IBD have been reported. In contrast to the protective role of the above three variants, the roles of these additional SNPs are either unspecified, protective, or aggravating. The variant rs10889677 is located in the 3′-untranslated domain of the IL-23R gene, leading to enhanced mRNA and protein production due to the loss of microRNA regulation. This dysregulation contributes to sustained IL-23R signaling, which is associated with increased susceptibility and contributes to the chronicity of IBD [106]. A recent meta-analysis of 41 studies confirmed that rs10889677 might be a risk factor for IBD, indicating that this variant is linked to CD risk in Caucasians but not in Asians. In contrast, it is associated with UC risk in Asians but not in Caucasians [82]. The variant rs7517847 shows varied results across IBD subtypes. It may reduce the risk of developing CD [83]. However, it has no association with UC risk, though it may be related to the presence of blood in stool and bowel movements in these patients [107]. Unspecified results are also noted for the rs1004819 variant, as some studies reported no association with UC [107], whereas others identified this polymorphism as the main disease-associated IL-23R variant in German CD patients [108]. Limited research data are available in other variants, such as rs2201841, rs11800503, rs7530511, and rs1884444 [109].

## 4. IL-23R-Based Treatment Options in IBD

Binding of IL-23 to IL-23R plays a critical and pivotal role in IBD progression, primarily through its effect on the differentiation and maintenance of Th17 cells in the IL-23/Th17 pathway, leading to the production of pro-inflammatory cytokines [13,110]. Due to the specific involvement of IL-23 in IBD pathophysiology, biopharmaceutical agents have recently been developed to more precisely reduce IBD-related inflammation compared to broad-spectrum immunosuppressants [111].

The first biopharmaceutical agent targeting the IL-23 pathway was ustekinumab. It targets the IL-12p40 subunit, which is shared by both IL-12 and IL-23, thereby affecting the activation of both interleukins. Ustekinumab is approved for treating moderate-to-severe CD and UC, even in patients who have not responded to previous biologics therapies [112,113]. However, its effects extend beyond the IL-23 pathway.

More recently, more selective therapeutic agents have been developed that specifically target the IL-23p19 subunit. One such agent is risankizumab, a high-affinity neutralizing antibody that binds to the IL23p19 subunit. It is a human IgG1 kappa antibody containing two mutations in the Fc region (Leu234Ala and Leu235Ala) that reduce Fc gamma receptor interactions [114]. Risankizumab inhibits IL-23 from interacting with the IL-23 receptor and thereby blocks the subsequent inflammatory signaling pathway, without affecting the IL-12 pathway [114]. Early clinical studies have shown effective results in both CD and UC, suggesting that risankizumab could become an important tool in the treatment of moderate-to-severe CD [115,116,117].

Another agent, mirikizumab, is a humanized monoclonal IgG4 antibody that selectively inhibits the IL23p19 subunit. It also blocks interactions with the IL-23R complex, reducing the release of pro-inflammatory cytokines that depend on IL-23 [118]. Recent studies have shown promising results in both CD and UC, indicating its potential effectiveness in the treatment of moderate to severe UC [119,120].

Furthermore, two additional antibodies, guselkumab and brazikumab, are currently under investigation. Both target the IL23p19 subunit [121]. Guselkumab is a humanized monoclonal IgG1 λ antibody that blocks the interactions with the IL-23R complex, thereby reducing cellular signaling and release of pro-inflammatory cytokines in the IL-23 axis [118]. Preliminary data suggest a potential benefit in the treatment of UC and CD [122,123]. Similar encouraging results have also been observed in studies for brazikumab [124].

In addition, scientific research is now targeting the IL23R directly. JNJ-77242113 is an investigational, first-in-class oral peptide designed to selectively block the IL-23 receptor in various immune-mediated inflammatory diseases involving IL-23 signaling pathways, such as psoriasis and IBD, without affecting IL-12 signaling. In a rat model of colitis induced by trinitrobenzene sulfonic acid, JNJ-77242113 attenuated disease parameters [125]. However, research on this oral peptide is limited to the animal experimental level.

While these treatments have been established in adult populations, their use in pediatric IBD remains largely limited to ustekinumab for refractory cases. This preference is based on its demonstrated safety profile comparable to adults, including consistent observations of weight improvement in children with growth failure and low adverse event rates. However, clinical experience with other IL-23 inhibitors in pediatric populations remains limited, highlighting a need for further investigation [126].

For female IBD patients of reproductive age, anti-IL23 biological factors are not contraindicated during pregnancy, in contrast with Janus kinase (JAK) inhibitors and methotrexate [127]. Specifically, clinical studies demonstrate that pregnancy outcomes with ustekinumab treatment show comparable effectiveness and safety profiles to those observed in the general population [128]. This favorable evidence positions IL-23 pathway inhibitors as a viable therapeutic option for managing IBD throughout pregnancy [128].

## 5. Discussion

The IL-23/IL-23R axis plays a central role in the immunopathogenesis of IBD, primarily through its regulation of the Th17 cell-mediated pro-inflammatory response. Genetic variations in the IL-23R gene significantly influence individual susceptibility to IBD, with specific polymorphisms such as rs11209026 (R381Q), rs76418789 (G149R), and rs41313262 (V362I) offering a protective effect, often by impairing receptor function, reducing stability, or altering signaling efficiency. In contrast, variants like rs10889677, which escape microRNA regulation, contribute to enhanced receptor expression and increased IBD risk. Other variants show population-specific or disease-subtype-specific effects, reflecting the complexity of genetic contributions to IBD. The prevalence of certain variants varies by ethnicity and region. For example, rs76418789 occurs more frequently in Asian populations, while rs11209026 is more common in Caucasian populations. Additional research on IL-23R gene variants is needed to better understand their distribution across different ethnic and regional groups. This knowledge would assist in developing population-specific treatment strategies.

From a therapeutic perspective, these insights have paved the way for the development of targeted biologics. Ustekinumab, initially developed to inhibit both IL-12 and IL-23 pathways, has proven effective in treating moderate-to-severe forms of CD and UC. The emergence of more selective inhibitors such as risankizumab, mirikizumab, guselkumab, and brazikumab, specifically blocking the IL-23p19 subunit, demonstrates the growing precision in IBD therapy. Moreover, novel strategies targeting the IL-23 receptor directly, such as the investigational oral peptide JNJ-77242113, represent a promising frontier in IBD treatment.

The initial clinical trials have demonstrated favorable safety profiles and positive long-term outcomes for IL-23R-targeted therapies. A recent phase 3b randomized trial confirmed that ustekinumab has a favorable risk–benefit profile in Crohn’s disease, with sustained clinical and endoscopic improvements through week 104 [129]. Furthermore, in patients with ulcerative colitis, mirikizumab has shown durable efficacy through 104 weeks, with no safety concerns [130].

Although IL-23R-targeted therapy has only been clinically available for a few years, their use is rapidly expanding mainly as a personalized medicine approach in refractory IBD. Emerging evidence suggests several biomarkers may predict treatment response. In a recent prospective pilot study, higher baseline serum IL-23 levels were associated with clinical and endoscopic response to ustekinumab in patients with CD, whereas fecal calprotectin levels were decreased in responders but not in non-responders [131]. Also, eosinophil reduction was highlighted as a marker for early response to ustekinumab in IBD, but not to adalimumab (anti-Tumor Necrosis Factor) and vedolizumab (anti-integrin) [132].

Most reported variants of the IL-23R gene exhibit a protective role in IBD. While gene editing technologies have not yet been explored as a potential therapeutic approach for IL-23R modification in IBD, related strategies have shown promise in autoimmune conditions. Notably, it has been demonstrated that antisense oligonucleotides have been shown to modify IL-23R splicing through exon nine skipping. This modification increases the production of an IL-23R isoform, which acts as a decoy receptor, attenuates IL-23 signaling and mimics the protective mechanism of the rs11209026 variant, thereby reducing the differentiation of pro-inflammatory Th17 cells [91]. Also, genetically engineered human adipose-delivered mesenchymal stem cells could express a recombinant IL-23 decoy receptor using a lentiviral vector, which could reduce the inflammatory responses in vitro [133]. These innovative approaches represent promising avenues for future targeted therapies in IBD.

## 6. Conclusions

Genetic and pharmacological findings underscore the IL-23/IL-23R pathway as a crucial determinant in the development and management of IBD. Protective variants in the IL-23R gene provide valuable insights into IBD pathophysiology, offering important clues about disease mechanisms and potential therapeutic targets. Future research into the functional consequences of specific IL-23R variants and ongoing therapeutic innovation will likely enhance personalized approaches to IBD care.

## Figures and Tables

**Figure 1 ijms-26-04775-f001:**
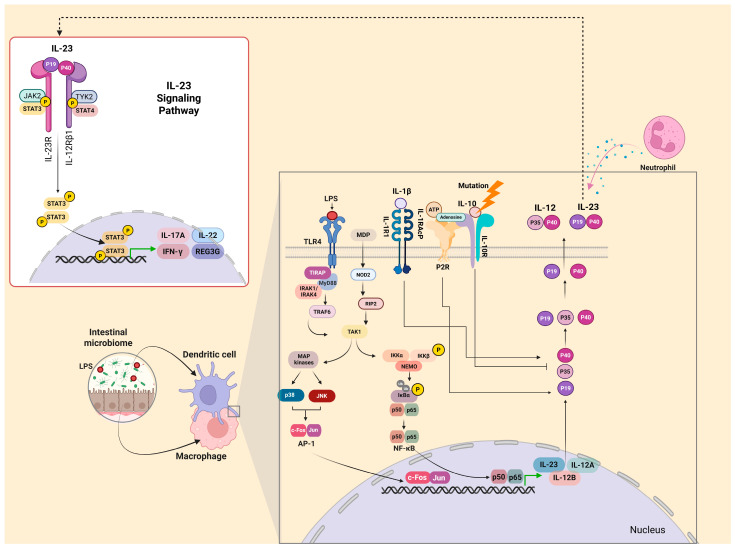
The IL-23 production and signaling pathway in the context of inflammatory bowel disease pathogenesis. Figure Legend: this figure illustrates the molecular mechanisms underlying IL-23 production and signaling and their central role in the immunopathogenesis of inflammatory bowel disease (IBD). The intestinal microbiome and its interaction with innate immune cells (e.g., macrophages and dendritic cells) are crucial in shaping immune responses. Microbe-derived signals stimulate these cells to produce cytokines, such as IL-23, that influence T cell polarization and perpetuate inflammation in the gut mucosa. IL-23 is produced primarily by antigen-presenting cells such as dendritic cells and macrophages upon stimulation by microbial components. Recognition of pathogen-associated molecular patterns (PAMPs), such as lipopolysaccharides (LPS) and muramyl dipeptide (MDP), occurs through pattern recognition receptors, including TLR4 and NOD2. This activation leads to the recruitment of adaptor molecules (e.g., MyD88, IRAK1/4, and RIP2) and the activation of downstream kinases such as TAK1. The signaling cascade culminates in the activation of transcription factors NF-κB and AP-1, which translocate to the nucleus and induce the expression of pro-inflammatory cytokines, including IL-12 (composed of p35 and p40 subunits) and IL-23 (composed of p19 and p40 subunits). The pathway also integrates input from the purinergic receptor P2R and inhibitory feedback mechanisms mediated by IL-10. However, genetic mutations affecting IL-10 or its receptor can impair this anti-inflammatory signaling, contributing to unchecked IL-23 expression. IL-23 secretion plays a pivotal role in sustaining inflammatory responses, particularly via its effects on Th17 cells and neutrophil recruitment. The IL-23 signaling pathway begins when IL-23 binds to its heterodimeric receptor complex, consisting of IL-23R and IL-12Rβ1. This engagement activates the associated Janus kinases (JAK2 and TYK2), leading to phosphorylation and activation of the transcription factors STAT3 and STAT4. Phosphorylated STAT3 dimerizes and translocates to the nucleus, where it promotes the transcription of pro-inflammatory effector molecules including IL-17A, IL-22, IFN-γ, and REG3G. These cytokines and antimicrobial peptides contribute to mucosal immune defense but also play pathogenic roles in the chronic inflammation observed in IBD. Created using https://BioRender.com (assessed on 14 April 2025). Abbreviations: IL-23, interleukin-23; JAK2, Janus kinase 2; STAT3/STAT4, signal transducer and activator of transcription 3/4; IL-23R, interleukin-23 receptor; IL-12Rβ1, interleukin-12 receptor β1 subunit; IL-17A, interleukin-17A; IL-22, interleukin-22; IFN-γ, interferon-gamma; REG3G, regenerating islet-derived protein 3 gamma; LPS, lipopolysaccharide; TLR4, Toll-like receptor 4; TIRAP, Toll/interleukin-1 receptor domain-containing adapter protein; NF-κB, nuclear factor kappa-light-chain-enhancer of activated B cells; p50/p65, subunits of NF-κB; NOD2, nucleotide-binding oligomerization domain-containing protein 2; TAK1, transforming growth factor beta-activated kinase 1; MDP, muramyl dipeptide; and P2R, purinergic receptor P2Y/P2X family.

**Figure 2 ijms-26-04775-f002:**
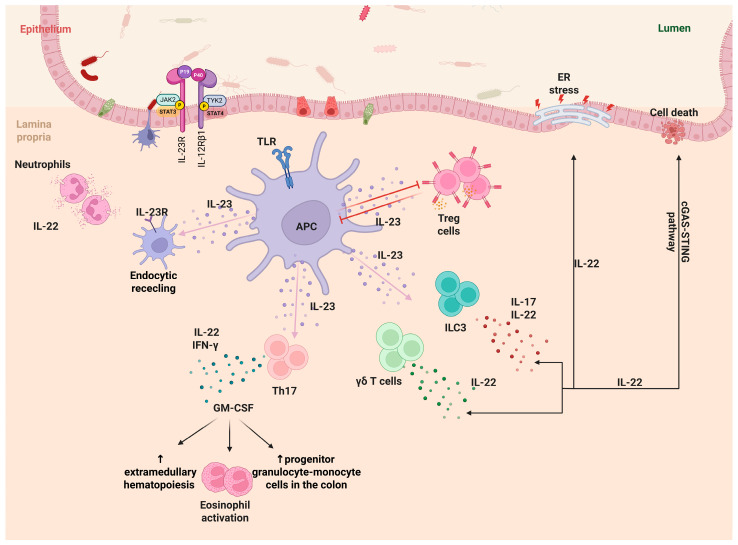
IL-23/IL-23R axis orchestrates innate and adaptive immune responses in the inflamed gut mucosa. Figure Legend: this figure depicts the complex immunological network modulated by IL-23 in the intestinal mucosa, highlighting its effects on both innate and adaptive immune compartments in the context of inflammatory bowel disease (IBD). Upon microbial recognition through Toll-like receptors (TLRs), antigen-presenting cells (APCs), such as dendritic cells and macrophages, secrete IL-23. The cytokine binds to the IL-23 receptor complex, composed of IL-23R and IL-12Rβ1, which is expressed on a variety of immune cells including Th17 cells, γδ T cells, innate lymphoid cells (ILC3), and neutrophils. Ligand-receptor binding activates the JAK2/TYK2-STAT3/STAT4 signaling cascade, resulting in the transcription of effector cytokines and inflammatory mediators. Th17 cells respond to IL-23 by producing IL-22 and IFN-γ, and notably granulocyte-macrophage colony-stimulating factor (GM-CSF), which promotes extramedullary hematopoiesis, eosinophil activation, and the recruitment of granulocyte-monocyte progenitors in the colon, contributing to mucosal inflammation. ILC3 and γδ T cells also respond to IL-23 by secreting IL-17 and IL-22, reinforcing barrier immunity and sustaining inflammation. Neutrophils, through IL-23R expression, engage in endocytic recycling processes and contribute to IL-22–mediated mucosal repair or damage, depending on the inflammatory context. Treg cells are shown to interact bidirectionally with the IL-23 axis. IL-23 exerts an inhibitory effect on Treg-mediated immunosuppression, whereas Tregs may attempt to counterbalance this activation via IL-10 and other regulatory mechanisms. The epithelial barrier is affected by IL-23-induced cytokines, particularly IL-22, which has dual roles: promoting epithelial proliferation and repair, while potentially triggering ER stress and cell death under chronic stimulation. This cellular stress contributes to the release of DNA into the cytoplasm and activation of the cGAS–STING pathway, which further amplifies type I interferon responses and intestinal inflammation. Created in https://BioRender.com (assessed on 14 April 2025). Abbreviations: Treg cells, regulatory T cells; ILC3, group 3 innate lymphoid cells; γδ T cells, gamma delta T cells; JAK2, Janus kinase 2; TYK2, tyrosine kinase 2; P, phosphorylation; STAT3, signal transducer and activator of transcription 3; STAT4, signal transducer and activator of transcription 4; p19, subunit of IL-23; P40, shared subunit of IL-12 and IL-23; IL-23R, interleukin-23 receptor; IL-12Rβ1, interleukin-12 receptor beta 1; TLR, Toll-like receptor; APC, antigen-presenting cell; IL-23, interleukin-23; Th17, T helper 17 cells; IFN-γ, interferon-gamma; ER stress, endoplasmic reticulum stress; cGAS-STING, cyclic GMP-AMP synthase–stimulator of interferon genes pathway; GM-CSF, granulocyte-macrophage colony-stimulating factor.

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
