# Peer review of "Targeting the IL-23 Receptor Gene: A Promising Approach in Inflammatory Bowel Disease Treatment"

_ijms, 2025, doi:10.3390/ijms26104775_

Round 1
Reviewer 1 Report
Comments and Suggestions for Authors
This review describes the function and potential role of IL-23 receptor in the treatment of inflammatory bowel diseases. In general, this review is well written and presented and has potential clinical significance in the development of novel bioagents by targeting IL-23 receptor signaling.
Previous studies have suggested that Interleukin (IL)-23 is a proinflammatory cytokine that can interact with other cytokines e.g., IL-12 and IL-17 to drive the development of chronic intestinal inflammation, therefore, the IL-23 and its receptor signaling have been implicated in the immunopathology of inflammatory bowel disease (IBD). This review first summarizes the effect and genetic mechanisms of IL-23 in modulating Th17 cell differentiation and IL-17 production. Their analysis suggested that synergistic interaction between IL-23 and IL-17 may be a critical driver of the chronic inflammation in the intestine and IBD. Based on published literature, the authors further showed that variants in IL-23 receptor gene may affect the risk for inflammation and IBD. Inhibition of IL-23 receptor function results in dampening the activation of proinflammatory pathways and ultimately reduces the risk for inflammation/IBD development. In general, this review is well written and presented. Th readers can benefit from this review for the understanding of IBD mechanisms, and it has a strong clinical significance in the development of novel bioagents by targeting IL-23 receptor signaling. Therefore, I suggest accepting it for publication on the present form.
Author Response
Thank you for your positive feedback on our manuscript. It is an honor to suggest accepting it on publication in its current form.
Reviewer 2 Report
Comments and Suggestions for Authors
Comment 1 :
Please make sure that the citations are properly formatted (they should be in square brackets, and they should correspond to the article you have cited). I would suggest using a program like EndNote, Zotero or Mendeley to organize your citations.
Comment 2:
I would suggest making the conclusions section shorter and consider including a discussion section, to better highlight the findings of your literature review.
Author Response
Comment 1: Please make sure that the citations are properly formatted (they should be in square brackets, and they should correspond to the article you have cited). I would suggest using a program like EndNote, Zotero or Mendeley to organize your citations.
Response to comment 1: All citations in the manuscript have been carefully reviewed and are now presented in square brackets, ensuring they correspond correctly to the referenced articles.
Comment 2: I would suggest making the conclusions section shorter and consider including a discussion section, to better highlight the findings of your literature review.
Response to comment 2: In the revised manuscript we have included a discussion section, and we have shortened the conclusion as suggested (lines 425-480).
Reviewer 3 Report
Comments and Suggestions for Authors
Ploutarchos et al. have provided a comprehensive review on the IL-23/Th17 pathway in inflammatory bowel disease, detailing how IL-23 promotes Th17 cell differentiation and inflammation with disease-specific effects on cytokine production. Their analysis of protective genetic variants in the IL23R gene (R381Q, G149R, V362I) that reduce IBD risk by impairing receptor function and downstream signaling provides a strong foundation for understanding this pathway as an important therapeutic target.
This is a well-structured review with thorough coverage of molecular mechanisms. However, I have several suggestions to further strengthen the manuscript:
- The authors note that the G149R variant is more prevalent in Asian populations. More research on the distribution of IL-23R gene variants across different ethnic and regional populations would be valuable, particularly addressing how these genetic differences might inform population-specific treatment strategies.
- The review focuses primarily on adult IBD. A section discussing the safety and efficacy of IL-23R-targeted therapy in pediatric IBD patients would enhance clinical relevance, as management approaches often differ between adult and pediatric populations.
- For female IBD patients of reproductive age, safety data regarding IL-23R-targeted therapies during pregnancy and lactation should be addressed. This information is critical for clinicians managing IBD in this patient population.
- The authors could expand on biomarkers that might predict which patients will respond favorably to IL-23R-targeted therapy. This would strengthen the section on personalized medicine approaches.
- Given the protective effects of certain IL-23R gene variants, it would be interesting to discuss whether any research is exploring gene editing technologies (e.g., CRISPR) to modify the IL-23R gene as a potential therapeutic approach.
- The manuscript would benefit from more detailed information on the duration of available safety follow-up data for these treatments and discussion of potential long-term risks.
- The figures are informative but complex. Consider adding a simplified schematic to illustrate the basic concepts of the IL-23/IL-23R signaling pathway for readers less familiar with molecular biology.
- The section on treatment options could be expanded to include more direct comparisons between IL-23-targeted therapies and other established IBD treatments, which would provide valuable context for clinical decision-making.
Author Response
Ploutarchos et al. have provided a comprehensive review on the IL-23/Th17 pathway in inflammatory bowel disease, detailing how IL-23 promotes Th17 cell differentiation and inflammation with disease-specific effects on cytokine production. Their analysis of protective genetic variants in the IL23R gene (R381Q, G149R, V362I) that reduce IBD risk by impairing receptor function and downstream signaling provides a strong foundation for understanding this pathway as an important therapeutic target.
This is a well-structured review with thorough coverage of molecular mechanisms. However, I have several suggestions to further strengthen the manuscript:
Response to Reviewer 3: Thank you for your positive feedback on our manuscript. We appreciate that you have provided constructive suggestions to strengthen and improve our manuscript. Below, we address each of your comments point by point.
Comment 1: The authors note that the G149R variant is more prevalent in Asian populations. More research on the distribution of IL-23R gene variants across different ethnic and regional populations would be valuable, particularly addressing how these genetic differences might inform population-specific treatment strategies.
Response to Comment 1: We have incorporated additional data from existing studies to better characterize the distribution of IL23R variants across different ethnic and regional populations. Specifically, for variant R381Q, we added supporting information in lines 280-282, for variant V362I, we included relevant data in lines 338-341, for variant rs10889677, the information was already included in our original manuscript (lines 365-366). We note that other variants in our study have not been as extensively investigated in IBD populations. In the discussion section (lines 434-439), we summarize the key findings on this topic and highlight the need for further research into the prevalence of IL23R gene variants across diverse populations.
Comment 2: The review focuses primarily on adult IBD. A section discussing the safety and efficacy of IL-23R-targeted therapy in pediatric IBD patients would enhance clinical relevance, as management approaches often differ between adult and pediatric populations.
Response to Comment 2: We have added a paragraph with the existing data for pediatric IBD patients and IL-23R-targeted therapy in lines 412-417.
Comment 3: For female IBD patients of reproductive age, safety data regarding IL-23R-targeted therapies during pregnancy and lactation should be addressed. This information is critical for clinicians managing IBD in this patient population.
Response to Comment 3: We have added a paragraph with the existing data for pregnancy and IL-23R-targeted therapy in lines 418-423.
Comment 4: The authors could expand on biomarkers that might predict which patients will respond favorably to IL-23R-targeted therapy. This would strengthen the section on personalized medicine approaches.
Response to Comment 4: We have incorporated a new paragraph discussing currently available data on predicted biomarkers for response to IL-23R-targeted therapies (lines 454-461).
Comment 5: Given the protective effects of certain IL-23R gene variants, it would be interesting to discuss whether any research is exploring gene editing technologies (e.g., CRISPR) to modify the IL-23R gene as a potential therapeutic approach.
Response to Comment 5: We have added a new discussion paragraph addressing current developments in gene editing technologies targeting the IL-23R pathway for inflammatory diseases (lines 462-472).
Comment 6: The manuscript would benefit from more detailed information on the duration of available safety follow-up data for these treatments and discussion of potential long-term risks.
Response to Comment 6: We have added a paragraph summarizing available safety follow-up data for IL-23R-targeted therapies (lines 449-454).
Comment 7: The figures are informative but complex. Consider adding a simplified schematic to illustrate the basic concepts of the IL-23/IL-23R signaling pathway for readers less familiar with molecular biology.
Response to Comment 7: In the revised manuscript, we have added a simplified schematic of the IL-23/IL-23R pathway to improve clarity for readers and have submitted it as a graphical abstract. This figure serves as an accessible introduction to the pathway's molecular and immunological concepts, as suggested by the reviewer.
Comment 8: The section on treatment options could be expanded to include more direct comparisons between IL-23-targeted therapies and other established IBD treatments, which would provide valuable context for clinical decision-making.
Response to Comment 8: We appreciate the reviewer's suggestion regarding comparative effectiveness of different IBD treatments. While we agree this represents an important clinical consideration, our current manuscript focuses specifically on the pathophysiology of IL-23R signaling in IBD and the therapeutic developments targeting this pathway. However, a comprehensive comparison of all established IBD treatments would indeed merit a separate systematic review, as it extends beyond our current scope.